# New Multicolor Tungstate-Molybdate Microphosphors as an Alternative to LED Components

**DOI:** 10.3390/ma14216608

**Published:** 2021-11-02

**Authors:** Justyna Czajka, Agata Szczeszak, Nina Kaczorowska, Stefan Lis

**Affiliations:** 1Faculty of Chemical Technology and Engineering, Bydgoszcz University of Science and Technology, Seminaryjna 3, 85-326 Bydgoszcz, Poland; justyna.czajka@pbs.edu.pl; 2Faculty of Chemistry, Adam Mickiewicz University, Uniwersytetu Poznańskiego 8, 61-614 Poznań, Poland; nina.kaczorowska@amu.edu.pl (N.K.); blis@amu.edu.pl (S.L.)

**Keywords:** luminescence, LED components, lanthanides, optical materials, tungstate–molybdate phosphors

## Abstract

Due to the ongoing need to create phosphors with the appropriate emission color for the production of light emitting diodes, we decided to synthesize a series of multicolour microphosphors with tunable visible emissions, depending on the composition of dopant ions. In this work, we investigated the structure, morphology, and luminescent properties of new molybdate–tungstate phosphors co-doped with Tb^3+^ and Eu^3+^ ions. The conventional high temperature solid state method was used to prepare a series of CaMo_y_W_1−y_O_4_:Eu^3+^_x_/Tb^3+^_1−x_ materials. In order to obtain phosphors with the most promising luminescent properties, the experiment was planned by taking into account the different composition of the matrix and the concentration of the particular dopant ions (Eu^3+^_x_/Tb^3+^_1−x_, x = 0.001, 0.003, 0.005, 0.007, 0.009). As a result, luminescent materials were obtained with a pure tetragonal crystal structure, the space group of *I*4_1_/a, confirmed by X-ray diffraction (XRD). The size and shape of the particles obtained from the materials were analyzed based on scanning electron microscopy images. Luminescence spectroscopy (excitation and emission spectra, decay lifetimes) was utilized to characterize the luminescence properties of the as-prepared phosphors. The color change of the emission from green-yellow to orange-red was confirmed using the 1931 Comission Internationale de l’Eclairage (CIE) chromaticity coordinates and color correlated temperature (CCT).

## 1. Introduction

Recently, many investigations have focused on the synthesis and characterization of molybdate [1,2,3,4,5] and tungstate [6,7,8,9,10] materials with a tetragonal scheelite structure [11,12,13] doped with trivalent lanthanide ions, Ln^3+^. Many different synthesis methods including sol-gel [5,14,15], sonochemical methods [16,17], hydrothermal methods [18,19], microwave assisted hydrothermal routes [20,21] as well as high temperature solid state methods [22,23,24], have been used to improve the structural and physicochemical properties of the synthesized molybdate/tungstate materials [25]. Economic aspects [7], technological importance [26], and possible applications have also been taken into account [7,27]. Inorganic luminescent materials doped with Ln^3+^ ions are well known as phosphors that are applied, for example, in solar cells [28,29], scintillators, light emitting diodes [30], lasers [31], bioimaging [14], and field emission displays [24]. Designing modern tunable luminescent materials with a wide range of applications has been a global trend for several years. Due to the rapid development of knowledge in the field of synthesis and properties of luminescent modern phosphors consisting of Ln^3+^ ions and tungstate or molybdate matrices, we decided to present new tunable phosphors based on mixed inorganic matrices. 

In this work, studies of the structure, morphology, and luminescence properties were carried out using different compositions of terbium, Tb^3+^, and europium, Eu^3+^, dopant ions, as well as with a different composition of the matrix (different W/Mo ratio) synthesized by the high temperature solid state method. Several techniques were used to characterize as-synthesized phosphors, including using X-ray powder diffraction, scanning electron microscopy, and photoluminescence spectroscopy, respectively. Under the selected excitation wavelength, Tb^3+^ and Eu^3+^ ions emit light with a color ranging from green through orange to red. Therefore, the synthesized phosphors have potential in an LED application. In this study, the correlated color temperature was calculated and the color emission was visualized on chromaticity diagrams and schemes.

The most recent articles describing materials for LEDs show that there remains a need for phosphors for the production of efficient and bright lighting LEDs [32] that are also produced based on a high temperature solid state reaction [33,34,35,36]. In fact, the literature reports extensive research on molybdate and tungstate matrices doped with Eu^3+^ [33,37,38] or Tb^3+^ ions [39,40]; however, the materials of the composition studied in the presented manuscript have not yet been investigated. Therefore, we present detailed studies of CaMo_y_W_1−y_O_4_:Eu^3+^_x_/Tb^3+^_1−x_ materials. The literature also provides information on tunable fluoride materials doped with Eu^3+^ and Tb^3+^ [41] or lithium borate glasses doped with Eu^3+^ and Tb^3+^ [42], which indicates the importance of the Eu^3+^/Tb^3+^ ion dopant system in the creation of novel lighting setups. 

The presented manuscript shows Eu^3+^/Tb^3+^—doped phosphors based on a mixed matrix, the luminescent properties of which are controlled by the matrix composition, as well as the number of dopant ions. As a result, the effects on changes in the correlated color temperature (CCT) and chromaticity parameters are analyzed in detail. According to our knowledge, such detailed data are not known in the literature.

## 2. Materials and Methods

CaMo_y_W_1−y_O_4_:Eu^3+^_x_/Tb^3+^_1−x_ phosphors were synthesized by the traditional solid state reaction method. The complete formula used for the required calculations is Ca_0.99_Mo_y_W_1−y_O_4_:Eu^3+^_x_/Tb^3+^_1−x_; however, for simplicity, the shorter version will be used in the text. In order to obtain materials exhibiting tunable intense emission, an experiment was planned regarding the matrix composition (CaMo_y_W_1−y_O_4_:Eu^3+^_x_/Tb^3+^_1−x_) and the concentration of dopant ions as follows: y = 0.1 − 0.9, x = 0.1 − 0.9. The starting materials were purchased from Stanford Materials (Lake Forest, CA, USA) (europium oxide (Eu_2_O_3_, 99.99%), terbium oxide (Tb_4_O_7_, 99.99%)), Sigma Aldrich (molybdenum oxide (MoO_3_, 99.9%), tungsten oxide (WO_3_, 99.9%)) as well as from POCh Gliwice (calcium carbonate (CaCO_3_, 99.5%)). In the first step, CaCO_3_, MoO_3,_ and WO_3_ were mixed at a 1:y:1−y molar ratio (y = 0.1 − 0.9), while in the second step, appropriate amounts of Eu_2_O_3_ and Tb_4_O_7_ were added and the total molar concentration of activator (Eu^3+^_x_/Tb^3+^_1−x_) ions was 1 mol.% (x = 0.001. 0.003. 0.005. 0.007. 0.009). The starting materials were ground thoroughly in an agate mortar for 1 h. Finally, the obtained homogeneous powders were placed in corundum crucibles and sintered in a muffle furnace at 900 °C for 4 h. After the calcination process, the samples were cooled down to room temperature. The structure of the final products was determined by X-ray powder diffraction (XRD) (BRUKER D8 Advance, Karlsruhe, Germany) using Cu-Kα radiation (λ = 0.1541874 nm), registered from 10° to 60° (2θ) with the step of 0.05°. Scanning electron microscopy (SEM) images were obtained with the use of a HITACHI SU3500 equipped with a BSE-3D detector (Tokyo, Japan), using an acceleration voltage of 10.0 kV. Excitation and emission spectra were measured on a Hitachi F-7000 fluorescence spectrofluorimeter (Tokyo, Japan) and corrected for the light source used (xenon lamp). The CIE 1931 chromaticity coordinates, x and y, were used to determine CCT using the McCamy equation [43].

## 3. Results

As a result of the solid state reaction, a series of tunable CaMo_y_W_1−y_O_4_ codoped with Eu^3+^ and Tb^3+^ ions was obtained. The representative composition of the matrix CaMo_0.5_W_0.5_O_4_ was selected and it was used in further studies as the most suitable for the introduction of dopant ions and for studying the dependency of Eu^3+^ and Tb^3+^ concentration on luminescence parameters. The influence of dopant ions on the structural properties of tungstate–molybdate matrices was analyzed based on XRD patterns for representative samples (Figure 1). The structure of the prepared materials can be indexed to a tetragonal crystal structure with the space group *I*4_1_/a of the scheelite type CaWO_4_ referenced by card number 96-900-9628. The crystal structure agreed well with card number 96-900-9633 and also confirmed the tetragonal structure CaMoO_4_. Based on the consistency of the diffractograms with the reference pattern, no other phases and impurities were detected. It was confirmed that the introduction of Tb^3+^ and Eu^3+^ dopants did not disturb the structure, thanks to the Ln^3+^ ions which were successfully incorporated into the host lattice [25,44]. In the described structures, hexavalent M^6+^ (M = W, Mo) atoms coordinate with four O^2−^ atoms and divalent Ca^2+^ atoms coordinate with eight O^2−^ atoms, respectively [22,45]. The similarity in the ionic radii between W^6+^ (0.42Å) and Mo^6+^ (0.41Å) confirmed that the positions of the diffraction peaks hardly changed [24], while the pointed diffraction peaks of high amplitude confirmed the high crystallinity of the synthesized luminescent materials [22]. In addition, the similarity of the ionic radii of Eu^3+^ (r = 1.07 Å for CN = 8) and Tb^3+^ (r = 1.04 Å for CN = 8) with the ionic radius of Ca^2+^ (r = 1.12 Å for CN = 8), and the purity of the samples, indicate that the lanthanide dopant ions are expected to substitute Ca^2+^ ions [6].

The scanning electron microscopy (SEM) images shown in Figure 2 present the morphology of CaMo_y_W_1−y_O_4_:Eu^3+^_x_/Tb^3+^_1−x_ luminescent materials synthesized by the high temperature solid state technique at 900 °C for 4 h. It is revealed that the samples have a topography typical for this method. Despite the tendency towards agglomeration [6,46], it was possible to estimate the average grain size, which covers the 0.5–12 μm range. To be precise, most of the samples exhibited a grain size in the range of 0.5–3 μm. The results of these estimates are shown in the form of size distribution in Figure 2.

Figure 3 shows the excitation spectra of CaMo_0.5_W_0.5_O_4_:Eu^3+^_x_/Tb^3+^_1−x_ measured at wavelengths corresponding to the most intense emission bands of Tb^3+^ (λ_em_ = 546 nm, ^5^D_4_ → ^7^F_5_) and Eu^3+^ (λ_em_ = 617 nm, assigned to ^5^D_0_ → ^7^F_2_). Both spectra present broad bands with maxima in the range of 200–350 nm, which is the result of the charge transfer (CT) phenomenon. These bands consist of the individual components such as the transitions O^2−^ → W^6+^, O^2−^ → Mo^6+^, O^2−^ → Tb^3+^, and O^2−^ → Eu^3+^. The intensity and broadness of the CT bands indicate the effectively activated emission of Eu^3+^ and Tb^3+^ ions through energy transfer between WO_4_^3−^ and MoO_4_^3−^ groups and activator ions, after CT between W^6+^ or Mo^6+^ and O^2−^ [24,47]. Moreover, in the excitation spectra, there are several weak absorption peaks of Eu^3+^ and Tb^3+^ (Figure 3a). The bands at 350, 358, 367, 377 and 486 nm are assigned to the 4f–5d transitions of Tb^3+^ from the ground level ^7^F_6_ to the ^7^D_2_, ^5^G_5_, ^5^G_6_, ^5^D_3_ and ^5^D_4_ excited levels of Tb^3+^, whereas the peaks at 376 and 393 nm are ascribed to the 4f–4f transitions of Eu^3+^ from the ground level ^7^F_0_ to the ^5^G_2_ and ^5^L_6_ excited levels of Eu^3+^, respectively. The excitation spectra monitored at λ_em_ = 617 nm (Figure 3b) consist of 360 nm (^7^F_0_ → ^5^D_4_), 375 nm (^7^F_0_ → ^5^G_2_), 380 nm (^7^F_0_ → ^5^G_3_), 393 nm (^7^F_0_ → ^5^L_6_), 415 nm (^7^F_0_ → ^5^D_3_), and 464 nm (^7^F_0_ → ^5^D_2_) peaks, typical for Eu^3+^. This means that we found the characteristic excitation from Eu^3+^ when monitoring the luminescence of Tb^3+^ (Figure 3a). On the other hand, when observing the Eu^3+^ luminescence, the characteristic peaks of Tb^3+^ are missing. This may indicate that the energy transfer from Eu^3+^ to Tb^3+^ is efficient.

Figure 4 shows the emission spectra of CaMo_0.5_W_0.5_O_4_:Eu^3+^_x_/Tb^3+^_1−x_ materials with excitation of 256 nm (a), 272 nm (b), and 394 nm (c). The emission spectra (Figure 4a,b) contain a wide range of bands from 400 to 600 nm assigned to the matrix emission, which indicates that the energy transfer from the host towards the dopant ions is not complete. Furthermore, it can be observed that its intensity decreases with an increasing amount of Eu^3+^ dopant ions. All samples show a strong emission at 617 nm and 546 nm upon 256 and 272 nm excitation. The emission peak at 617 nm is assigned to the ^5^D_0_ → ^7^F_2_ electric dipole transition of Eu^3+^, while the band at 546 nm is ascribed to the ^5^D_4_ → ^7^F_5_ of Tb^3+^. Other peaks are assigned at 490 nm (^5^D_4_ → ^7^F_6_ of Tb^3+^), 589 nm (^5^D_4_ → ^7^F_5_, Tb^3+^), 593 nm (^5^D_0_ → ^7^F_1_, Eu^3+^), 657 nm (^5^D_0_ → ^7^F_3_, Eu^3+^) and 705 nm (^5^D_0_ → ^7^F_4_, Eu^3+^). Apart from Eu^3+^, under the excitation of 394 nm, there are also emission bands assigned to the Tb^3+^ ion. This may indicate effective energy transfer from Eu^3+^ to Tb^3+^ (Figure 5) [48].

The decay curves of the Tb^3+^ transitions at 490 nm, 546 nm, and 589 nm in CaMo_0.5_W_0.5_O_4_:Eu^3+^_x_/Tb^3+^_1−x_ phosphors were measured under the 256 nm excitation wavelength as represented in Figure 6a–c, respectively. The decay curves of Eu^3+^ were also measured under the 256 nm excitation wavelength for the emissions at 617 nm, 656 nm, and 705 nm, which correspond to the ^5^D_0_ → ^7^F_2_, ^5^D_0_ → ^7^F_3_, and ^5^D_0_ → ^7^F_4_ transitions, respectively. All curves were fitted by one-exponential decay. To identify and analyze more detailed ET from Eu^3+^ to Tb^3+^, we recorded decay curves under λ_ex_ = 394 nm (Figure 6g). The slightly longer lifetime of Tb^3+^ (λ_em_ = 546 nm) and shorter lifetime time of Eu^3+^ (λ_em_ = 617 nm) may indicate energy migration from Eu^3+^ to Tb^3+^.

Commission International de L’Eclairage (CIE) chromaticity coordinates for CaMo_y_W_1−y_O_4_:Eu^3+^_x_/Tb^3+^_1−x_ materials with different doping concentrations of Tb^3+^ and Eu^3+^ were calculated based on the corresponding emission spectra. The CIE chromaticity diagram, chromaticity coordinates, and CCT values for CaMo_y_W_1−y_O_4_:Eu^3+^_x_/Tb^3+^_1−x_ excited at 256 nm are shown in Figure 7. Correlated color temperature (CCT) values were calculated using the McCamy equation [43]. We observed color shifts from yellow-green (Figure 7 sample 1, 5, Figure 8) towards the orange-red region (Figure 7 sample 44, 45, Figure 8) with an increase in the Eu^3+^ doping concentration. The CCT values are in the range of 2062–6842 K under λ_ex_ = 256 nm. Figure 7 shows the results shown for all samples series. We observed similar values for the samples No. 31 (x = 0.3803, y = 0.3771, 4007 K), 33 (x = 0.4261, y = 0.3811, 2998 K), and 34 (x = 0.4425, y = 0.3839, 2735 K) with the main colorimetric data of the LED illuminator V2 (x = 0.3781, y = 0.3775, 4070 K), B2 (x = 0.4357, y = 0.4012, 2988 K), and B1 (x = 0.4560, y = 0.4078, 2773 K), respectively, recommended by CIE [49]. Figure 8 shows pictures of CaMo_y_W_1−y_O_4_:Eu^3+^_x_/Tb^3+^_1−x_ under UV light in Eppendorf tubes and glass tubes.

## 4. Conclusions

In summary, multicolor molybdate–tungstate microphosphors co-doped with Tb^3+^ and Eu^3+^ were synthesized successfully using the high temperature solid state method. XRD analysis shows that the synthesized phosphors are single-phase with a tetragonal *I*4_1_/a structure. The excitation spectra show a charge transfer band between 200 and 350 nm which effectively activates the emissions of Eu^3+^ and Tb^3+^ ions through energy transfer between WO_4_^3−^ and MoO_4_^3−^ groups and activator ions, after CT between W^6+^ or Mo^6+^ and O^2−^. Scanning electron microscopy (SEM) images presented a strong tendency for phosphors to agglomerate, which is specific to the high temperature solid state synthesis method we used. At 265 nm excitation, the CaMo_y_W_1−y_O_4_:Eu^3+^_x_/Tb^3+^_1−x_ microphosphors show color shifts from yellow-green to orange-red with an increasing doping concentration of Eu^3+^. The intense emission upon UV light excitation indicates that the new multicolored tungstate–molybdate microphosphors can be used as an alternative to LED components. Detailed studies of Eu^3+^/Tb^3+^-doped phosphors based on the mixed molybdate–tungstate matrix, whose luminescent properties were controlled by the composition of the matrix and the number of doping ions, showed that the obtained values of correlated color temperature (CCT) and chromaticity parameters were similar to those recommended by CIE. In addition, the use of a relatively small number of doping ions, compared to others described in the literature, allows for intense emission of the tested phosphors, which is important from an economic point of view.

## Figures and Tables

**Figure 1 materials-14-06608-f001:**
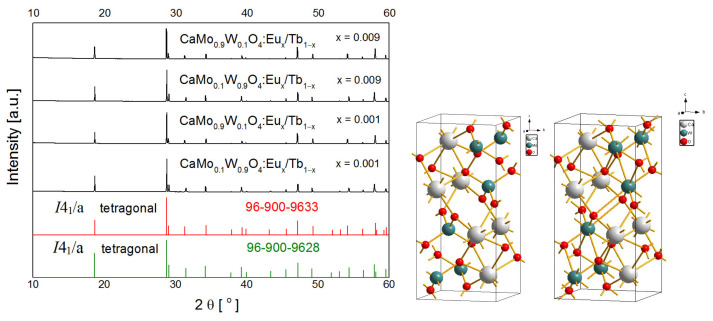
XRD patterns of CaMo_0.9_W_0.1_O_4_ and CaMo_0.1_W_0.9_O_4_ materials doped with Eu^3+^_x_/Tb^3+^_1−x_ (x = 0.001, x = 0.009), synthesized at 900 °C for 4 h.

**Figure 2 materials-14-06608-f002:**
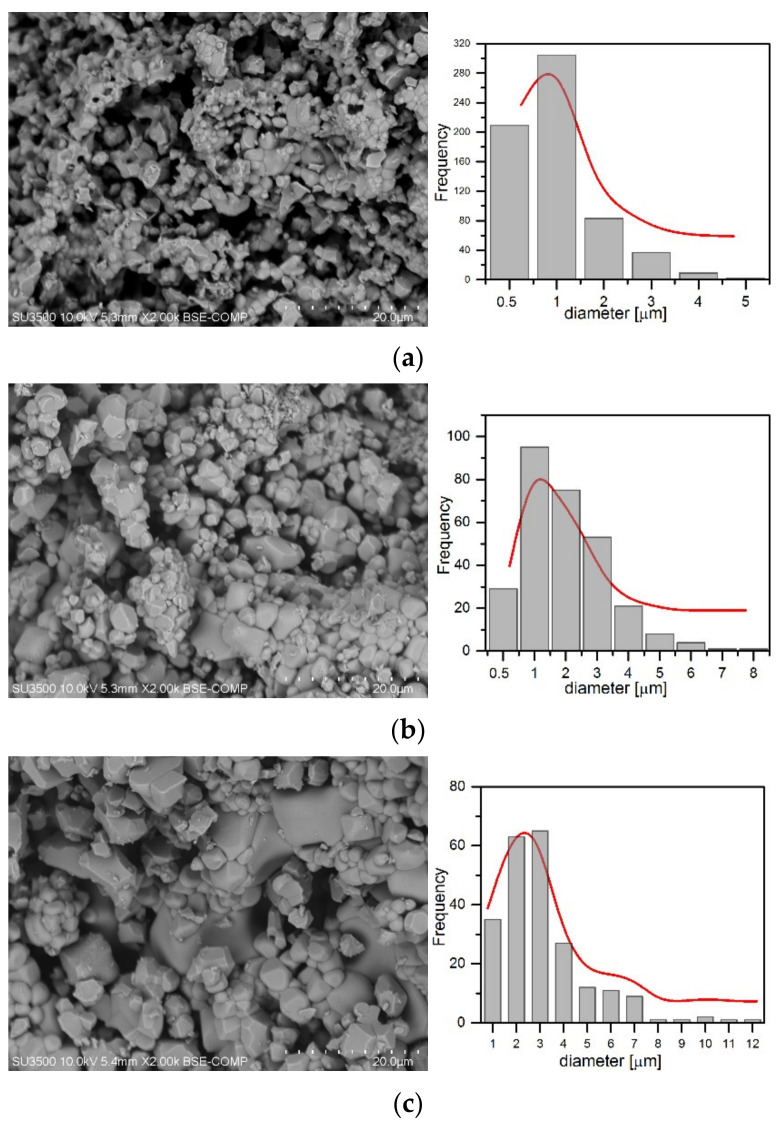
SEM images and histograms of samples (**a**) CaMo_0.5_W_0.5_O_4_:Eu^3+^_0.005_/Tb^3+^_0.005_, (**b**) CaMo_0.9_W_0.1_O_4_:Eu^3+^_0.009_/Tb^3+^_0.001_, (**c**) CaMo_0.1_W_0.9_O_4_:Eu^3+^_0.001_/Tb^3+^_0.009_.

**Figure 3 materials-14-06608-f003:**
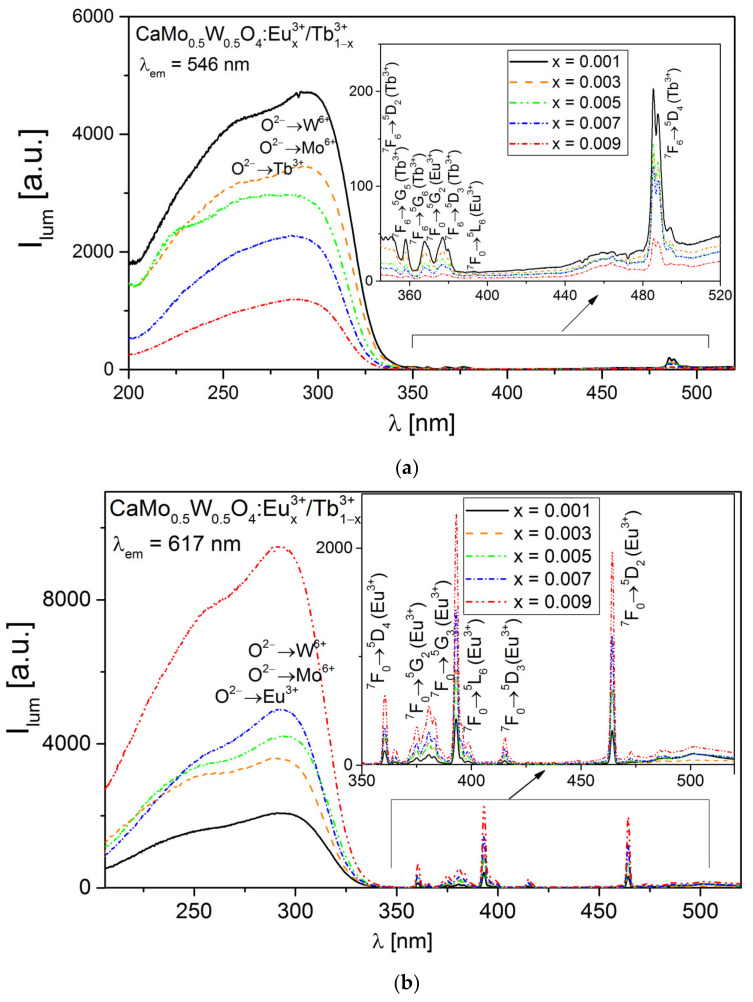
Excitation spectra of CaMo_0.5_W_0.5_O_4_:Eu^3+^_x_/Tb^3+^_1−x_ phosphors monitored at a wavelength of (**a**) λ_em_ = 546 nm and (**b**) λ_em_ = 617 nm. The inserts magnify the bands in the range from 350 to 520 nm.

**Figure 4 materials-14-06608-f004:**
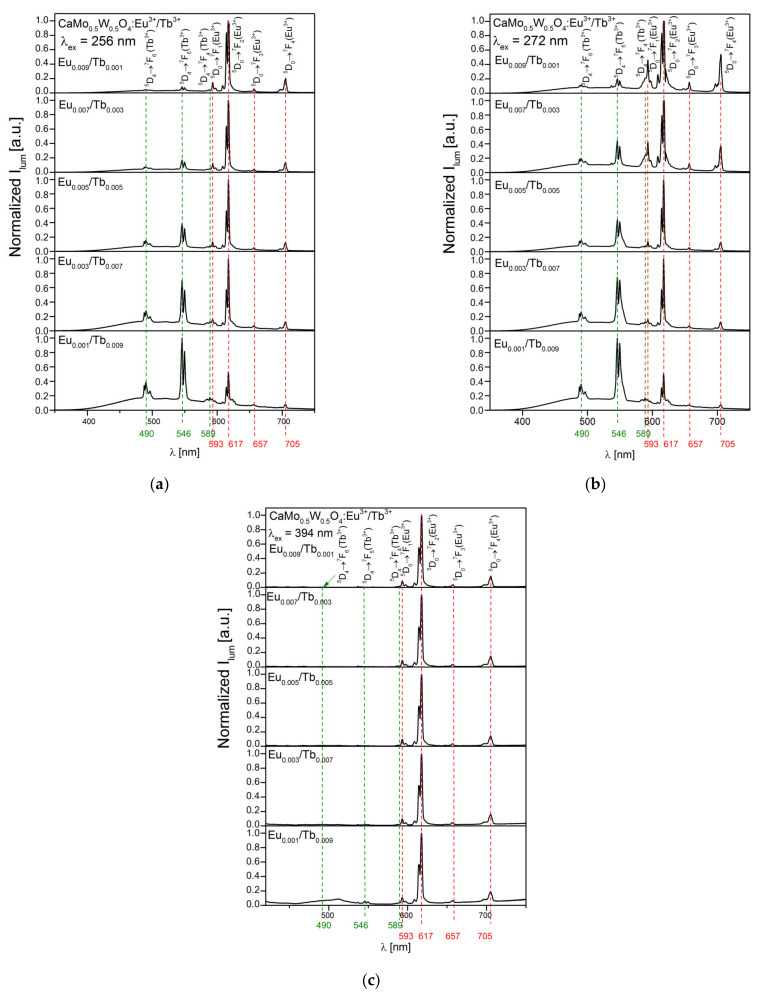
Emission spectra of CaMo_0.5_W_0.5_O_4_:Eu^3+^_x_/Tb^3+^_1−x_ phosphors (**a**) λ_ex_ = 256 nm, (**b**) λ_ex_ = 272 nm, (**c**) λ_ex_ = 394 nm.

**Figure 5 materials-14-06608-f005:**
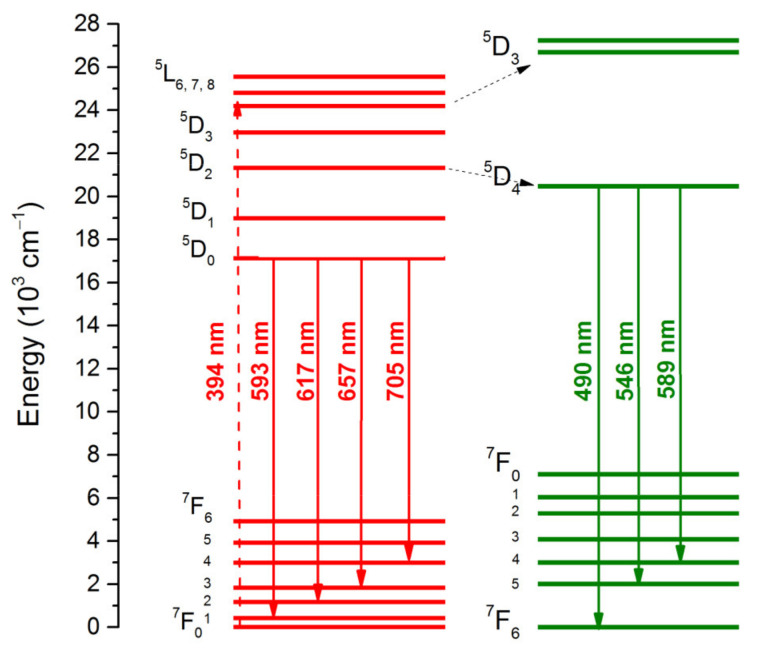
Energy level scheme diagram showing ET from Eu^3+^ to Tb^3+^.

**Figure 6 materials-14-06608-f006:**
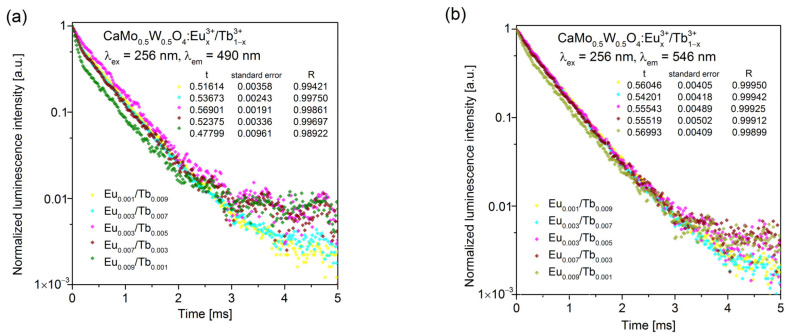
Luminescence decay curves for CaMo_y_W_1−y_O_4_:Eu^3+^_x_/Tb^3+^_1−x_ (**a**) λ_ex_ = 256 nm, λ_em_ = 490 nm, (**b**) λ_ex_ = 256 nm, λ_em_ = 546 nm, (**c**) λ_ex_ = 256 nm, λ_em_ = 589 nm, (**d**) λ_ex_ = 256 nm, λ_em_ = 617 nm, (**e**) λ_ex_ = 256 nm, λ_em_ = 656 nm, (**f**) λ_ex_ = 256 nm, λ_em_ = 705 nm, (**g**) λ_ex_ = 394 nm, λ_em_ = 546 nm, λ_em_ = 617 nm.

**Figure 7 materials-14-06608-f007:**
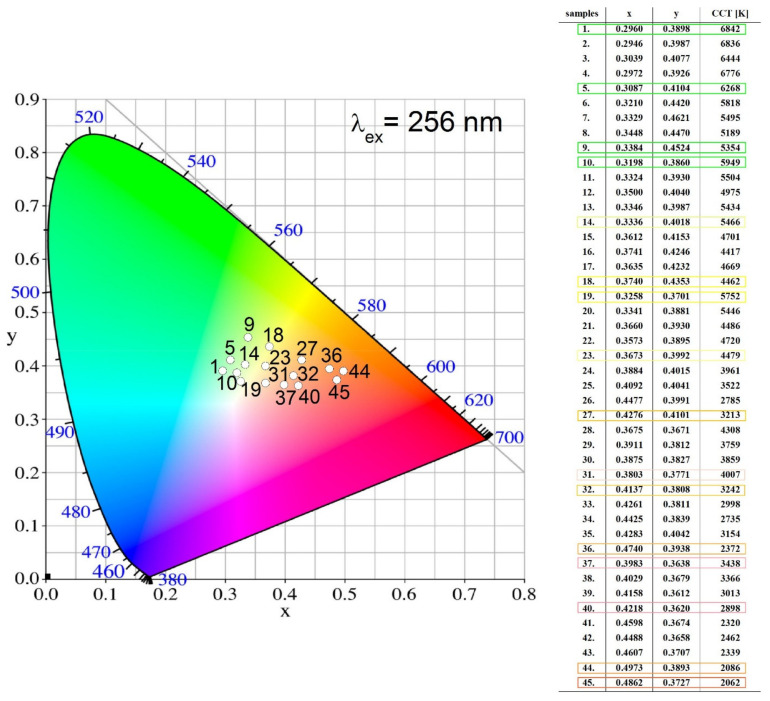
CIE chromaticity diagram, chromaticity coordinates, and correlated color temperature (CCT) of all phosphors of the CaMo_y_W_1−y_O_4_:Eu^3+^_x_/Tb^3+^_1−x_ series excited at 256 nm: series 1–9: Eu^3+^_0.001_/Tb^3+^_0.009_Mo_(0.1–0.9)_W_(0.9–0.1)_, series 10–18: Eu^3+^_0.003_/Tb^3+^_0.007_Mo_(0.1–0.9)_W_(0.9–0.1)_, series 19–27: Eu^3+^_0.005_/Tb^3+^_0.005_Mo_(0.1–0.9)_W_(0.9–0.1)_, series 28–36: Eu^3+^_0.007_/Tb^3+^_0.003_Mo_(0.1–0.9)_W_(0.9–0.1)_, series 37–45: Eu^3+^_0.009_/Tb^3+^_0.001_Mo_(0.1–0.9)_W_(0.9–0.1)_.

**Figure 8 materials-14-06608-f008:**
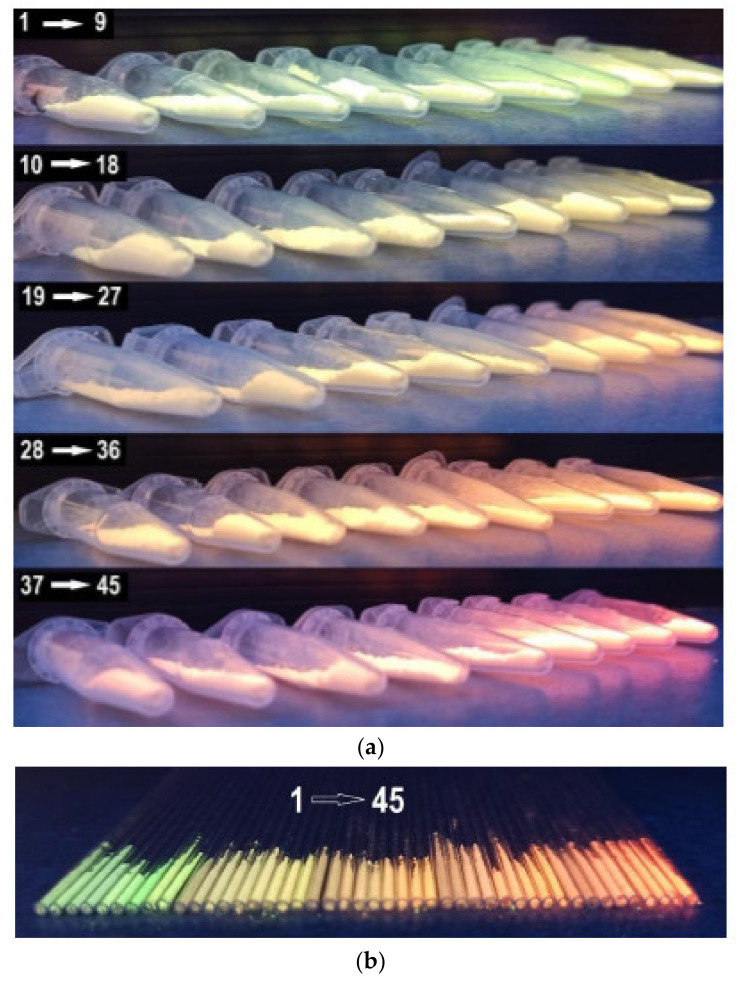
Images of CaMo_y_W_1−y_O_4_:Eu^3+^_x_/Tb^3+^_1−x_ under UV in (**a**) Eppendorf tubes and (**b**) glass tubes.

## Data Availability

Data sharing not applicable.

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
