# Peer review of "New Multicolor Tungstate-Molybdate Microphosphors as an Alternative to LED Components"

_materials, 2021, doi:10.3390/ma14216608_

Round 1
Reviewer 1 Report
This article ‘New multicolour tungstate-molybdate microphosphors as an alternative to LED components’ shows the synthesis and characteristic of a series of CaMoyW1-yO4Eu3+x/Tb3+1-x materials. The work is very symmetric and complete, but some key issues must be revised after considering the publication in the journal:
- The authors should point out what is novel for a design of these materials, and what is state-of-the-art materials for LED application.
- In line 82 on page 2, the authors directly selected y =0.5 in further study. Can the authors state a reason?
- The fonts in all figures are very small.
- It is possible to show a simple property of an LED device by using a material?
Author Response
Please see the attachment with the answers

Reviewer 2 Report
This work fills in some gaps in the previously published work on tungstate/molybenate phosphors. But in order to specify what is new. the authors need to greatly improve their introduction and background section. 1) The known structure of these phosphors must be introduced including a figure with the illustration of the unit cell.
2) The specific differences between the phosphors and the phosphors already published in the literature need to be clearly explained. The introduction and background section needs to be much longer and it should include a statement something like: " the literature (ref) reports tungstate molybenate phosphors doped with ? but ?? has not been studied. We therefore present details of ??. It is useful/important to know the results in this paper because...."
3) The insets in Figure 2 are much too small. Please provide insets as separate figure. Please explain how grain size was determined given that agglomerates are observed.
4) Emission spectra in Figure 4 are much too small. Please reorganize this figure to increase the size of each spectrum
5) The conclusions should reflect the achievement according to the revised introduction and background section
Round 2
Reviewer 1 Report
Because the authors have responded to all comments and revised the manuscript as well, it can be accepted as the present.
Reviewer 2 Report
Corrections look good